# Comparison of Contrast Enhanced Magnetic Resonance Angiography to Computed Tomography in Detecting Pulmonary Arteriovenous Malformations

**DOI:** 10.3390/jcm9113662

**Published:** 2020-11-14

**Authors:** Daniel A.F. van den Heuvel, Marco C. Post, Ward Koot, Johannes C. Kelder, Hendrik W. van Es, Repke J. Snijder, Jan-Albert Vos, Johannes J. Mager

**Affiliations:** 1Department of Radiology, St. Antonius Hospital, 3435 CM Nieuwegein, The Netherlands; w.koot@antoniusziekenhuis.nl (W.K.); h.es@antoniusziekenhuis.nl (H.W.v.E.); j.a.vos@antoniusziekenhuis.nl (J.-A.V.); 2Department of Cardiology, St Antonius Hospital, 3435 CM Nieuwegein, The Netherlands; m.post@antoniusziekenhuis.nl (M.C.P.); keld01@antoniusziekenhuis.nl (J.C.K.); 3Department of Cardiology, University Medical Center Utrecht, 3584 CX Utrecht, The Netherlands; 4Department of Pulmonology, St Antonius Hospital, 3435 CM Nieuwegein, The Netherlands; r.snijder@antoniusziekenhuis.nl (R.J.S.); j.mager@antoniusziekenhuis.nl (J.J.M.)

**Keywords:** pulmonary arteriovenous malformation, contrast enhanced magnetic resonance angiography, hereditary hemorrhagic telangiectasia

## Abstract

Background: Computed tomography (CT) is considered the imaging modality of choice to diagnose pulmonary arteriovenous malformations PAVMs. The drawback of this technique is that it requires ionizing radiation. Magnetic resonance (MR) imaging does not have the limitation, but little is known about the performance of MR compared to CT for the detection of PAVMs. The aim of this study is to investigate the sensitivity of contrast-enhanced MR angiography (CE-MRA) in the detection of PAVMs with feeding artery diameters (FAD) > 2 mm. Methods: Patients with a grade 2 or 3 shunt on screening transthoracic contrast echocardiography (TTCE) were asked to participate. Included patients underwent chest CT and CE-MRA. CT was considered the reference standard. CT and CE-MRA scans were anonymized and assessed for the presence of PAVMs with FAD > 2 mm by one and two readers respectively. Data analysis was performed on per patient and per PAVM basis. Results: Fifty-three patients were included. 105 PAVMs were detected on CT, 45 with a FAD ≥ 2 mm. In per patient analysis, sensitivity and specificity of CE-MRA were 92% and 97% respectively for reader 1 and 92% and 62% for reader 2. Negative and positive predictive value (NPV/PPV) were 93% and 96% for R1 and 90% and 67% for R2. In per PAVM analysis, sensitivity, specificity, NPV and PPV were 96%, 99%, 100% and 86% for R1 and 93%, 96%, 100% and 56% for R2, respectively. Conclusions: CE-MRA has excellent sensitivity and NPV for detection of PAVMs with FAD ≥ 2 mm and can therefore be used to detect these PAVMs. We are hopeful that future advancements in CE-MRA technology will reduce false positive rates and allow for more broad use of CE-MRA in PAVM diagnosis and management.

## 1. Introduction

Pulmonary arteriovenous malformations (PAVM) are vascular malformations frequently encountered in patients with hereditary hemorrhagic telangiectasia (HHT), also known as Rendu–Osler–Weber disease. In this type of vascular malformation, the normal capillary bed is absent and there is a direct connection between the pulmonary artery and vein. As a result, unfiltered and unsaturated blood enters the systemic circulation, which may cause hypoxemia and forms a potential conduit for paradoxical emboli. The latter may lead to serious clinical complications; the most feared are cerebral infarctions or cerebral abscesses [1,2]. Hence, it is important that these vascular malformations are detected and treated before they cause irreversible damage. In the 2011 International Guidelines for the Diagnosis and Management of HHT, it is recommended that all PAVMs with a feeding artery diameter (FAD) of 3 mm and, if feasible, >2 mm are treated [3]. Computed tomography (CT) is considered the imaging modality of choice to diagnose these PAVMs [3,4]. The drawback of this technique is that it requires ionizing radiation. This is important to realize because in this relatively young population CT investigations are used during long-term follow-up to assess PAVM persistence after embolotherapy and growth of untreated PAVMs [5,6,7,8,9,10]. Hanneman et al., reported an average of four chest CT scans (range 0–20) per patient and that 11% of their study cohort received a cumulative effective dose of >100 mSv at which patients are considered to be at risk for radiation induced harm [11]. Advancements in low-dose and ultra-low-dose (submillisievert) CT imaging have significantly lowered radiation burden [12]. This facilitates a more liberate use of CT in PAVM management, probably without increasing the radiation risks. However, the ‘as low as reasonably achievable’ principle still applies in radiology and technical developments in magnetic resonance imaging (MRI) offer the possibility of dynamic and functional imaging of PAVMs without radiation burden. It is therefore, interesting to investigate the role of MRI in PAVM management. A first prerequisite is that MRI can reliably detect PAVMs. Therefore, the aim of this study was to investigate the sensitivity of contrast-enhanced MR angiography (CE-MRA) compared to CT in the detection of PAVMs with a FAD > 2 mm.

## 2. Materials and Methods

All subjects gave their informed consent for inclusion before they participated in the study. The study was conducted in accordance with the Declaration of Helsinki and was approved by the institutional review board of our hospital (MEC-U NL39415.100.12). 

### 2.1. Patient Population

Consecutive patients presenting at the HHT outpatient clinic (a tertiary referral center for HHT), suspected having untreated PAVMs based on a moderate or severe shunt (grade 2 or 3) diagnosed by transthoracic contrast echocardiography (TTCE), were asked to participate in the study. As part of our protocol, all patients with a moderate or severe shunt on TTCE undergo a chest CT scan to evaluate the presence of PAVMs. After signing informed consent, they also underwent a CE-MRA of the pulmonary arteries. 

### 2.2. Trans Thoracic Contrast Echocardiography Protocol

A detailed description of the TTCE protocol can be found in the paper by Velthuis et al. [13]. The contrast agent, consisted of 8 mL of saline 0.9%, 1 mL of room air and 1 mL of blood, was administration through the right antecubital vein. The number of microbubbles in the left side of the heart was counted on one still frame. A moderate (grade 2) or large (grade 3) shunt was present if 30–100 or >100 microbubbles appeared in the left ventricle, respectively. All shunts visualized in the left atrium through a pulmonary vein or after at least four cardiac cycles were classified as a pulmonary shunt.

### 2.3. CT Acquisition Protocol

All CT scans were performed on a 256-slice scanner (Philips Brilliance iCT, Best, The Netherlands). Scan direction was from the lung bases to apices and in a single breath hold. Scan parameters were 100 kV, reference mAs 120, collimation 128 × 0.625, pitch 0.68. Images were reconstructed at 1 mm slice thickness. In case of a previous embolization procedure a contrast enhanced CT scan was performed. Scan parameters were identical to nonenhanced CT scans. Scan delay after administration of 80 mL Iobitridol (Xenetix^®^ 300, Guerbet Laboratories, Roissy, France) was 30 s. A 40 mL saline flush was administered at the beginning of the scan.

### 2.4. CE-MRA Acquisition Protocol

CE-MRA was performed at 1.5 T (Philips Achieva, Best, The Netherlands) using a five-element cardiac coil. Time-resolved CE-MRA acquisition was with a T1 weighted fast field echo (FFE), repetition/echo time of 2.9/1.5 and a flip angle of 30 degrees. Field of view was 320 × 345 with a 200 × 200 matrix, 230 slices with slice oversampling factor of 1.1, slice thickness reconstruction 1.5 mm and 0.57 × 0.57 × 1.5 voxel. The resulting sagittal 3D scan was repeated three times: noncontrast, arterial phase and venous phase. Contrast agent used is Dotarem (Guerbet 0.5 mmol/m; 30 mL flow 2.5 mL/sec followed by saline flush of 20 mL).

### 2.5. CT and CE-MRA Assessment

There were two readers in this study. Reader 1 (R1) with >10 years of experience in chest and interventional radiology was also involved in the development of the MRA protocol. Reader 2 (R2) was a resident in radiology and reviewed five learning cases in a session with R1 prior to study assessment. First, all CT scans were reviewed by R1. The CE-MRAs were independently and randomly reviewed by R1 and R2. To minimize recall bias, the time interval between the CT and the CE-MRA reviewing sessions was > four weeks for R1. All scans were assessed for the presence and location of PAVMs. Both readers scored the location per segment on CE-MRA to enable adequate correlation between the CT and MRA. For this the Jackson and Huber classification was used [14]. R1 measured the feeding artery diameter on CT within 1 cm of the sac. The 2011 guidelines recommend embolization of PAVMs with feeding artery diameter of 3 mm and if technically feasible also of PAVMs with feeding artery diameters of 2 mm [3]. In the current study, all PAVMs with a FAD of >2 mm were assessed. For the assessment of the CE-MRA the sagittal source images as well as axial 1 mm reconstructions and the sagittal and axial maximum intensity projections were used (Figure 1). 

### 2.6. Statistical Analysis

CT was considered the reference standard and the CE-MRA the index test. Per-patient and per-PAVM analysis were performed for both readers. The per-patient analysis was limited to distinguishing between the presence or absence of PAVMs with a feeding artery diameter of ≥2 mm in an individual. The per-PAVM analysis was focused on finding and correlating all PAVMs per segment. Sensitivity, specificity and negative predictive values (NPV) and positive predictive value (PPV) were calculated for the per-patient analysis. For the per-PAVM analysis the same test statistics were calculated taking into account that for each patient 18 segments were screened. With this approach it was also possible to determine the specificity and NPV for the per-PAVM analysis. Statistical analysis was performed with R version 3.6.3 [15].

## 3. Results

Fifty-five consecutive HHT patients with a pulmonary shunt (grade > 1) on TTCE were asked to participate. Two patients declined and therefore 53 patients were included in this study representing 105 PAVMs. Forty-five PAVMs had a feeding artery diameter of ≥2 mm (42.9%). Out of these, 11 PAVMs had a feeding artery diameter of ≥3 mm. The 45 PAVMs with a FAD ≥ 2 mm were present in 24 patients (45.3%). Patients’ demographics are shown in Table 1.

### 3.1. Per-Patient Analysis

With the threshold set at a diameter of the feeding artery of ≥2 mm, R1 correctly diagnosed PAVMs in 22 patients based on CE-MRA. The absence of PAVMs was correctly diagnosed in 28 patients; there were two false negative cases and one false positive case (sensitivity 92% (73–99%) and specificity 97% (82–100%)). Reader 2 also correctly diagnosed PAVMs in 22 patients. The absence of PAVMs was correctly diagnosed in 18. Similar to R1, R2 had two false negative cases, but 11 false positive cases (sensitivity 92% (73–99%) and specificity 62% (42–79%)). Both readers correctly diagnosed PAVMs in all patients when the threshold of ≥ 3 mm was applied. Results are presented in Table 2 and Table 3.

### 3.2. Per-PAVM Analysis 

One hundred and five PAVMs were identified in a total of 954 segments (53 patients × 18 segments per patient) on CT, of which 45 PAVMs had a feeding artery diameter of ≥2 mm. R1 correctly diagnosed 43 of 45 PAVMs and had seven false positives (sensitivity 96% (85–99%) and specificity 99% (98–100%)) on CE-MRA. R2 identified 42 of 45 PAVMs and had 33 false positives (sensitivity 93% (82–99%) and specificity 96% (95–97%)). As in the per-patient analysis, all PAVMs were identified by both readers when the threshold was set at ≥3 mm. Results are presented in Table 3 and Table 4.

## 4. Discussion

In this study, CE-MRA showed a high sensitivity and negative predictive value for the diagnosis of PAVMs with a FAD ≥ 2 mm in patients with PAVMs and a TTCE confirmed significant pulmonary shunt. PAVMs with a FAD ≥ 2 mm were diagnosed by chest CT in 45% of patients with at least a moderate pulmonary shunt on TTCE. PAVMs with a FAD ≥ 2 mm were found in 43% using CE-MRA, with a sensitivity of 92% and a NPV between 90 and 93%. 

Earlier, we have shown that, in the presence of a moderate or severe shunt on TTCE, the chest CT is negative for PAVMs in 55% and 8%, respectively [13]. Chest CT has an important role in the detection and follow-up of PAVMs in HHT. Its high spatial resolution enables imaging of feeding arteries, sacs and draining veins and thereby is able to accurately diagnose and assess treatment effect of PAVMs. However, the use of chest CT comes at the expense of ionizing radiation, which is an important disadvantage in this cohort of HHT patients, who are generally young and are likely to require several follow-up CT scans during their lifetime. In addition, metal artifacts hamper evaluation of PAVMs with CT after coil embolization. MRA is not limited by radiation burden and also seems to perform well with respect to the evaluation of PAVM persistence.

In the last two decades, a handful of studies have been published reporting on the feasibility of MRI in the management of PAVMs. In 2008, Schneider et al. reported a study including 203 HHT patients evaluating CE-MRA as a screening procedure for the detection of PAVMs [16]. They showed that CE-MRA is more sensitive in detecting PAVMs compared to pulmonary angiography (PA), which was the reference standard. Overall, there were 40 patients identified in whom 119 PAVMs were found with screening and follow-up using CE-MRA. Interestingly, with PA only 92 PAVMs were found (77%). As pointed out by the authors, there were important limitations in this study. First, patients with a negative CE-MRA were not compared to the reference standard. Second, there was no comparison to chest CT as the most important diagnostic tool in current algorithms. A third limitation was that the lower threshold for detection was defined as a sac diameter of at least 5 mm instead of feeding artery diameter ≥ 2 mm which is advocated in the guidelines [3]. 

Although not the subject in our study, MRA could also have a role in the follow-up of embolized PAVMs. Several studies have demonstrated the feasibility of (CE)-MRA for detecting PAVM persistence [17,18,19,20]. Kawai et al. used time-resolved CE-MRA to assess PAVMs and correlated with the recanalization rate on chest CT and PA [18]. The sensitivity (96%) and specificity (96%) were excellent, with good interobserver agreement. Important limitations of this study were the low number of included PAVMs (*n* = 28) and that 3 PAVMs could not be imaged by CE-MRA because of field of view limitation. For the detection of PAVM perfusion after embolotherapy, Hamamoto et al. investigated the feasibility of time-spatial labeling inversion pulse MRA (time-SLIP MRA or arterial spin labeling), a cinematic MRI method in which the inversion pulse facilitates the determination of blood flow without a contrast medium [20]. In 11 patients, 38 PAVMs were imaged with the time-SLIP MRA technique that was compared to PA as reference standard. The sensitivity and specificity of time-SLIP MRA for detecting perfusion after embolization were both 100%, with an excellent inter-observer agreement. Although very promising, time-SLIP MRA is also limited by the relatively small field of view. In this small study, four PAVMs could not be imaged because of this. 

MRA is a very promising imaging method for PAVMs but there are limitations. Our study and previous studies have shown that (CE)-MRA is a feasible method to detect PAVMs. It performs well compared to chest CT in the detection of PAVMs with a FAD ≥ 2 mm. However, the number of false positives needs to be improved. The experience of the operator seems to play an important role in this, as there is a large difference between both operators in the number of false positives. With respect to the detection of perfusion of embolized PAVMs, MRA seems to perform superior compared to CT because it is less influenced by the presence of micro coils [17,18]. This is attributable to the fact that micro coils are of platinum material which has few paramagnetic effects. MRA could therefore be feasible for the postembolization follow-up as well. As mentioned above, with CE-MRA and time-SLIP MRA, it is possible that the field of view is not large enough to cover the whole lung. The time-SLIP MRA technique often requires targeted imaging and thus knowledge of the location of the PAVM of interest. This means that either extra contrast administration during the MRA investigation or a chest CT scan prior to the MRA investigation will be necessary. Considering these limitations, the relative high costs and limited availability of MR imaging, it may be too early to use (CE)-MRA for routine evaluation of PAVMs. 

Our study has several limitations. MRA acquisitions were performed on a 1.5 T system. With a 3 T system, higher signal-to-noise ratios can be achieved. This could result in superior image quality based on a higher spatial and temporal resolution. In addition, it enables a broader coverage of the lung reducing the risk of false negative readings. Unfortunately, a 3 T system was not available during the study. Furthermore, there was a difference in experience in reading CE-MRA investigations between both readers. This study showed that image interpretation is reader-dependent, mainly resulting in false positives by the less experienced reader. Reader 1 has both a large experience in chest radiology and PAVM embolotherapy. He represents the level of experience that can be expected in an expert center. Reader 2, although being trained in an expert center, has less experience and could be regarded as representing nonexpert centers. HHT guidelines recommend that PAVM screening, diagnosis and treatment should be performed in expert centers. Furthermore, both readers were aware of the fact that the subjects had a positive TTCE and thus a high chance of having PAVMs with a FAD ≥ 2 mm. This could have induced a review bias lowering the threshold for diagnosing PAVMs. Finally, a limitation was the relatively small number of patients in this study. However, it is largest cohort of consecutive patients investigated so far representing a reasonably number of PAVMs with a FAD ≥ 2 mm.

## 5. Conclusions

CE-MRA seems to be a feasible method for the detection of PAVMs with a FAD ≥ 2 mm in HHT patients initially screened with TTCE. We are hopeful that future advancements in CE-MRA technology will reduce false positive rates and allow for more broad use of CE-MRA in PAVM diagnosis and management.

## Figures and Tables

**Figure 1 jcm-09-03662-f001:**
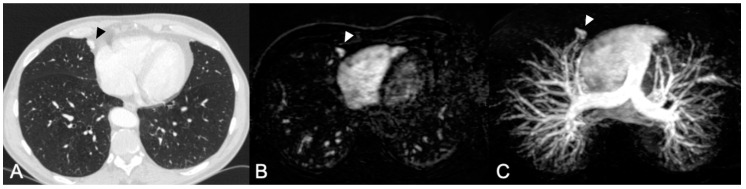
(**A**) Computed tomography (CT) scan 1.5 mm slice thickness with an inconspicuous PAVM (pulmonary arteriovenous malformation) in the medial segment of the middle lobe (arrow). (**B**,**C**) CE-MRA (contrast enhanced magnetic resonance angiography) axial reconstruction and axial MIP (maximum intensity projection) demonstrating the same PAVM.

**Table 1 jcm-09-03662-t001:** Baseline patient characteristics.

Total	53
Female	29 (55)
Male	24 (45)
Age (y)	47 ± 16
HHT	52 (98)
*Eng*	36 (68)
*Alk*	5 (9)
*SMAD 4*	3 (6)
Unknown ***	8 (15)
Idiopathic	1 (2)
PAVMs per patient	
0	9 (17)
1	15 (28)
2–5	19 (36)
>5	10 (19)

Data are presented as number, (%) and mean ± SD; HHT = hereditary hemorrhagic telangiectasia; PAVM = pulmonary arteriovenous malformation; * diagnosis only based on Curacao criteria.

**Table 2 jcm-09-03662-t002:** Contingency table of Reader 1 and 2 per-patient analysis.

Reader 1	CT +	CT −	Total
CE-MRA +	22	1	23
CE-MRA −	2	28	30
Total	24	29	53
Reader 2	CT +	CT −	Total
CE-MRA +	22	11	33
CE-MRA −	2	18	20
Total	24	29	53

**Table 3 jcm-09-03662-t003:** Sensitivity, Specificity, NPV and PPV of CE-MRA.

	Per Patient Analysis		Per PAVM Analysis	
	CE-MRA Reader 1	CE-MRA Reader 2	CE-MRA Reader 1	CE-MRA Reader 2
Sensitivity	92 (73–99)	92 (73–99)	96 (85–99)	93 (82–99)
Specificity	97 (82–100)	62 (42–79)	99 (98–100)	96 (95–97)
NPV	93 (78–99)	90 (68–99)	100 (99–100)	100 (99–100)
PPV	96 (78–100)	67 (48–82)	86 (73–94)	56 (44–67)

Data are in percentages, data in parenthesis are 95% confidence intervals. PAVM = pulmonary arteriovenous malformation, CE-MRA = contrast enhanced magnetic resonance angiography, NPV = negative predictive value, PPV = positive predictive value.

**Table 4 jcm-09-03662-t004:** Contingency table of Reader 1 and 2 per-PAVM analysis (18-segment model).

Reader 1	CT +	CT −	Total
CE-MRA +	43	7	50
CE-MRA −	2	902	904
Total	45	909	954
Reader 2	CT +	CT −	Total
CE-MRA +	42	33	75
CE-MRA −	3	876	879
Total	45	909	954

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
