# Peer review of "Comparison of Contrast Enhanced Magnetic Resonance Angiography to Computed Tomography in Detecting Pulmonary Arteriovenous Malformations"

_jcm, 2020, doi:10.3390/jcm9113662_

Round 1

Reviewer 1 Report

This is a single center, (retrospective), non-randomized study comparing CE-MRA with CT in patients with PAVM. The report does not give much new information. The authors do not mention the rather high cost and not general accessibility of MR at all institutions. The study was performed with 1.5T system which is not up-to-date,should have been 3T. The population examined was selected as all patients had a positive TTCE which might induce a significant review bias.

In Abstract: R1 and R2 should be written in full and explained.

Materials and Methods: P2, l. 55 "little is known about the performance of MR..... - not true; many publications available by perusing the literature. The new guidelines from CVIR should also have been referred to.

Results: P4, l. 133 Table 3 should come before Table 4 in the text and "Table 3" should changed to "Table 4" and "Table 4" to "Table 3".

In Table 1 "Sex" could be deleated. Was the diagnoses in 8 patients with "no mutations"  listed under HHT, based on clinical findings and according to the Curacao criteria? 9 patients with 0 PAVMs had shunt >grade 1? Was there any explanation for these significant shunts? Please comment.

There are many false positives and the experience of the operator seems to play an important role

Discussion: P5, l. 150 "in patients with HHT" should be "with PAVM".

P5, l. 154 - 161: The authors seem bias'ed in favor of MR: The population does not seem especially young as the mean age in this study is 47 y. Follow-up will usually be by TTCE and if negative no further controls by CT are needed. Low-dose CT is not considered in the discussion. It is mentioned that metal artifacts hamper evaluation of PAVMs with CT after embolization, but nothing about the effect on MR.

P5, l. 164: ref 12 is NOT by Schneider, byt by Ohno et al 

References: Not up-to-date, eg #12 old MR study. Several newer studies missing.

Reviewer 2 Report

I feel there are a number of issues which need clarification and/or further development:

  1. The use of “treatable” in the title and throughout the manuscript is problematic for a number of reasons. PAVMs that are “treated” in a patient are actually determined at the time of pulmonary angiography and not at the time of the non-invasive imaging; not all “treatable” PAVMs based on CT findings end up being treated. Since the authors are not comparing CE-MRA to pulmonary angiography, I don’t think that using “treatable” in this manuscript makes sense. In reality, the authors are comparing CE-MRA with CT to identify PAVMs with a FAD>= 2mm. In my mind, a better comparison would be to compare CE-MRA not only the CT technology used in this group of patients (which is likely a different technology than the CT technology used in earlier publications) but also to findings and “treatment” decisions at time of pulmonary angiography. Since I am pretty sure that each of their patients who had a “treatable” PAVM on “the imaging modality of choice” had a pulmonary angiogram, I bet the authors have this data and I such an analysis would be a better contribution to the field. Even with angiography data, since the term “treatable” is subjective and confusing, I think changing the primary focus to FDA >= 2mm and secondary focus to PAVMS that were treated at time of pulmonary angiography would be a better analysis. In addition, references to imaging modality of choice are from 9 years ago; I suspect practice has changed since then.
  2. 53 patients is a small data set for an analysis like this.
  3. Analysis of radiation risk from chest CT scans is based on a publication > 6 years old. More recent CT imaging acquisition and processing technologies have likely reduced radiation exposure and effects (for instance, a moderate dose, non-contrast Chest CT at my institution exposes a subject to no more radiation than the subject is exposed to over a full year from background environmental exposure.)
  4. Methods
    1. Patient Population
  1. TTCE: what type of “contrast” used?
  2. What is definition of “moderate or severe” shunt and “grade 2 or 3”? reference here would be helpful as well.
  3. What is definition of a “treatable” PAVM? (see above)

2.4 CT and CE-MRA assessment: need more information in order or publish and more importantly to understand what was done, especially by R2. What R1 did is pretty well described, but since not much is written about R2, as written I have to assume that R2 did not do much. I think less interpretation bias if CT and CE-MRA were randomly review instead of sequentially. I know what the “2011 guidelines” are, but the novice might not, therefore need to clarify here. While good here to define ”treatable” PAVMs, a bit late in the manuscript to see the definition (also see my earlier comments about “treatable”).

Figure 1: interesting, but I do not find very helpful, especially when focus is on FAD >= 2mm and no where else is there a definition of MIP nor any discussion of “suppressed mediastinal fat”.

  1. Discussion: more time than I think is warranted for discussion of a technology not used in this study: time-SLIP MRA

I am concerned that last sentence of the manuscript about “imaging quality should improve….” May be wishful thinking and I would end on a more realistic and/or cautious note.

Round 2

Reviewer 2 Report

The revised and resubmitted manuscript is a significant improvement over the initial submission and is now worthy and acceptable for publication with only a few minor edits for clarity of communication.

  1. TITLE: Given clarity in the paper and probable space constraints, the Title is OK, but a better title might be:  Comparison of Contrast Enhanced Magnetic Resonance Angiography to Computed Tomography in Detecting Pulmonary Arteriovenous Malformations".

2.  Lines 23-25: "Patients suspected of having PAVMs with FAD >2 mm based on transthoracic contrast echocardiography grade  >1 were asked to participate. "  I am pretty sure that the work by this group and others regarding the predictive use and value of TTCE for PAVMS was for finding a PAVM on CT that would yield the finding of a PAVM on pulmonary angiography that was treate; that the focus was on eventual treatment and not on FAD.  If I am correct, then need to craft a sentence that conveys the accurate intent.  IF I am incorrect, then the sentence is OK.  Either way, her in Abstract, how about simply stating: "Patients with a grade 2 or 3 shunt on screening transthoracic contrast echocardiography were asked to participate. " ?

3. Lines 35-36: this sentence is confusing: "Image quality should improve to reduce the substantial false positive rates for more broad applicability of CE-MRA in PAVM management." How about something like: "We are hopeful that future advancements in CE-MRA technology will reduce false positive rates and allow for more broad use of CE-MRA in PAVM diagnosis and management"

4. Line 48: "In the HHT guidelines..." is too non-specific despite providing a reference. Suggest: "In the 2011 International Guidelines for the Diagnosis and Management of HHT,...." (this also helps to socialize these important guidelines.)

5. Lines 56-64: nice.

6.  Line 182: Typo.  Suggest change  "However, the use of chest CT comes at the However, it comes at" to "However, the use of chest CT comes at"

7. Line 218: Typo.  Suggest change "(re)perfusion" to "reperfusion"

8. Lines 248-249: this sentence is confusing: "Image quality should improve to reduce the false positive rates before CE-MRA can be more broadly applicable in PAVM evaluation." How about something like: "We are hopeful that future advancements in CE-MRA technology will reduce false positive rates and allow for more broad use of CE-MRA in PAVM diagnosis and management"
